# Isolated Cognitive Decline in Neurologically Stable Patients with Multiple Sclerosis

**DOI:** 10.3390/diagnostics11030464

**Published:** 2021-03-07

**Authors:** Jiri Motyl, Lucie Friedova, Manuela Vaneckova, Jan Krasensky, Balazs Lorincz, Jana Blahova Dusankova, Michaela Andelova, Tom A. Fuchs, Eva Kubala Havrdova, Ralph H. B. Benedict, Dana Horakova, Tomas Uher

**Affiliations:** 1Department of Neurology and Center of Clinical Neuroscience, First Faculty of Medicine and General University Hospital, Charles University in Prague, 128 21 Prague, Czech Republic; jiri.motyl@vfn.cz (J.M.); lucie.friedova@vfn.cz (L.F.); balazs.lorincz@vfn.cz (B.L.); blahova.dusankova@gmail.com (J.B.D.); michaela.andelova@vfn.cz (M.A.); eva.kubalahavrdova@vfn.cz (E.K.H.); dana.horakova@vfn.cz (D.H.); 2Department of Radiology, First Faculty of Medicine and General University Hospital in Prague, Charles University in Prague, 128 08 Prague, Czech Republic; manuela.vaneckova@vfn.cz (M.V.); jan.krasensky@vfn.cz (J.K.); 3Buffalo Neuroimaging Analysis Center, Department of Neurology, Jacobs School of Medicine and Biomedical Sciences, University at Buffalo, State University of New York, Buffalo, NY 14203, USA; tomfuchs@buffalo.edu (T.A.F.); benedict@buffalo.edu (R.H.B.B.)

**Keywords:** cognitive decline, isolated cognitive decline, cognition, disease activity monitoring, relapse, disability, MRI, multiple sclerosis

## Abstract

(1) Background: Cognitive deterioration is an important marker of disease activity in multiple sclerosis (MS). It is vital to detect cognitive decline as soon as possible. Cognitive deterioration can take the form of isolated cognitive decline (ICD) with no other clinical signs of disease progression present. (2) Methods: We investigated 1091 MS patients from the longitudinal GQ (Grant Quantitative) study, assessing their radiological, neurological, and neuropsychological data. Additionally, the confirmatory analysis was conducted. Clinical disease activity was defined as the presence of new relapse or disability worsening. MRI activity was defined as the presence of new or enlarged T2 lesions on brain MRI. (3) Results: Overall, 6.4% of patients experienced cognitive decline and 4.0% experienced ICD without corresponding clinical activity. The vast majority of cognitively worsening patients showed concomitant progression in other neurological and radiologic measures. There were no differences in disease severity between completely stable patients and cognitively worsening patients but with normal cognition at baseline. (4) Conclusions: Only a small proportion of MS patients experience ICD over short-term follow-up. Patients with severe MS are more prone to cognitive decline; however, patients with normal cognitive performance and mild MS might benefit from the early detection of cognitive decline the most.

## 1. Introduction

Cognitive deterioration is an important marker of disease activity in multiple sclerosis (MS) which may occur without worsening of physical disability [1,2]. It is important to focus not only on the benchmark of cognitive impairment but, for the purpose of the disease activity monitoring, to also detect cognitive decline even before the threshold into cognitive impairment is crossed [3]. To detect such early cognitive changes, it was recommended to monitor cognitive processing speed on an annual basis using sensitive yet easily administered tests such as the Symbol Digit Modalities Test [2]. Cognitive changes often accompany other indices of disease progression [2,3,4] but can also occur in otherwise neurologically or radiologically stable patients [1,2,5,6,7]. Such cognitive changes may provide clinicians information on disease progression, she/he would not recognize otherwise.

Cognitive changes independent of neurological or radiological activity may occur as isolated cognitive relapses with incomplete or partial recovery [7] or changes taking a form of a gradual cognitive decline [1,2,3]. The concept of isolated cognitive decline (ICD), and its usefulness for disease progression monitoring, is still a matter of ongoing discussion and research [3,8,9]. The underlying processes leading to cognitive decline are often difficult to appreciate with conventional MRI markers as utilized in common clinical practice [10,11,12,13]. 

Currently, it is not clear, what is the proportion of patients with ICD in a real-world setting. Previous studies addressing this question were done only on small samples, followed mostly only isolated cognitive relapses, or chose strict concept and inclusion criteria [7,14]. 

The aim of this study was to investigate the proportion of MS patients with ICD. The secondary aim was to describe characteristics of patients with increased risk of ICD.

## 2. Materials and Methods

### 2.1. Study Population 

In this study, we investigated a large sample of patients from the Grant Quantitative (GQ) study over a two-year follow-up, revisiting our previous observational data [15,16]. The GQ study was a prospective observational study investigating the application of a comprehensive battery of clinical and paraclinical measures to evaluate MS progression in routine clinical practice. The inclusion criteria were as follows: MS confirmed by MRI and cerebrospinal fluid examination [17], Czech native speaker, participation in a brain MRI volumetric assessment program, and age 18 or more. The exclusion criteria were signs and symptoms suggestive of a disease other than MS and a serious psychiatric disorder. Enrolment into the GQ study started in June 2012. 

In the main analysis, we included 1091 participants (out of the original 1253) with available data on the Expanded Disability Status Scale (EDSS), relapses, demographic information, and Symbol Digit Modalities Test (SDMT) performed at baseline and at Month 12. At baseline, 90.8% of patients were diagnosed with relapsing-remitting and 9.2% with secondary progressive MS. The confirmatory analysis was conducted on the second year of follow-up between the Month 12 and Month 24 time points, with the aim to verify the primary findings. In the confirmatory analysis, we included 1060 participants fulfilling analogical criteria (see the study design in Figure 1). 

The GQ study was approved by the Medical Ethics Committee of the General University Hospital in Prague and First Faculty of Medicine, Charles University in Prague. 

### 2.2. Magnetic Resonance Image Acquisition and Analysis 

The original GQ study used MRI scans performed within three months before or after neuropsychological assessment at baseline, Month 12, and Month 24. A standardized protocol was performed using a single 1.5-Tesla scanner (Gyroscan; Philips Medical Systems, Best, The Netherlands) in the Department of Radiology at the General University Hospital in Prague, Czechia. Axial brain images were acquired using fast fluid-attenuated inversion recovery (FLAIR) and T1-weighted three-dimensional fast field echo images. Semiautomated subtraction image methodology (ScanView software) [18] was used to identify radiological worsening (i.e., active T2 lesions on brain MRI), defined as new (≥3 mm in diameter) or enlarging (≥50% of the original size) lesions on FLAIR scans, occurred during the preceding 12 months. Detection of active T2 lesions was performed by clinical researchers under neuroradiologist’s supervision. BPF was calculated as the total brain parenchymal volume divided by the intracranial brain volume (calculated as the sum of the total brain parenchymal volume, the total intra-ventricular cerebrospinal fluid volume, and the subarachnoidal cerebrospinal fluid volume). The corpus callosum normalized volume was calculated in a similar way, by dividing the corpus callosum volume by intracranial brain volume. 

### 2.3. Neuropsychological Assessment 

We tested all participants with the Czech validated version of the Brief International Cognitive Assessment for MS test battery (BICAMS) and the Czech language Rao adaptation of Paced Auditory Serial Addition Test-3 sec. (PASAT-3). In the follow-up time points, alternate forms were used [19,20,21]. In this analysis, we included results from the Symbol Digit Modalities Test (SDMT) oral version, processing speed assessment [22], as one of the tests most suitable for reliable annual screening in the clinical environment thanks to its psychometric properties and a short time of administration [1]. For the assessment of depressive symptoms, we used the Czech version of the Beck Depression Inventory (BDI) [23].

### 2.4. Neurological Assessment 

Clinical monitoring included regular visits every three months with an assessment of the Expanded Disability Status Scale (EDSS) score. In addition, patients had acute visits in case of suspected relapsing activity defined as patient-reported symptoms or objectively observed signs typical of an acute inflammatory demyelinating event in the central nervous system with a duration of at least 24 h, in the absence of fever or infection. EDSS worsening was defined as any increase of EDSS score (i.e., EDSS change > 0) over a one-year follow-up. Patients with relapse were treated with high-dose steroids. Visits within 30 days after relapsing activity or high-steroid treatment were excluded from the analysis.

### 2.5. Symptoms Evaluation

#### 2.5.1. Cognitive Impairment and Decline

We used the benchmark of −1.5 SD for cross-sectional cognitive impairment in SDMT, using the regression-based norms of 134 healthy controls adjusted for age and education [19]. Patients were evaluated as having cognitive impairment when scoring outside the normal range [19,20]. 

Cognitive decline was evaluated using the Lord–Novick reliable change index (RCI) [24,25], comparable to other RCI methodologies [26,27]. This procedure allowed us to minimalize Type I Error by accounting for test-retest reliability and other standard error of measurement issues. 

Firstly, we calculated a baseline true score estimation, accounting for test-retest reliability and regression to mean (Formula 1, i.e., Formula 7.2.3 in Lord and Novick [24], where E[τ] = baseline true score estimation, $r_{xx\prime }$ = test-retest reliability, X = SDMT total score, $M_{x}$ = SDMT population mean score)
(1)$$\;E\left[\tau \right]=r_{xx\prime }\times X\;+\;\left(1\;-\;r_{xx\prime }\right)\times M_{x}].\;$$

Afterward, we calculated the difference between the observed retest score and the estimated baseline true score. We evaluated an individual’s clinical course of cognition as a confirmed cognitive decline when the calculated difference was wider than the 95% Confidence Interval (Formula 2, where ±z = 1.96) of Standard Error of Prediction (Formula 3, i.e., Formula 3.8.5 in Lord and Novick [24], where $\sigma_{\mathrm{pred}}$ = Standard Error of Prediction, $\sigma_{x}$ = Standard Deviation of SDMT population mean score, $r_{xx\prime }^{2}$ = test-retest reliability square root). As test-retest reliability we used SDMT intraclass correlation of 0.85 [28].
(2)$$\pm CI_{95\%}=\pm z\times \sigma_{\mathrm{pred}}$$
(3)$$\sigma_{\mathrm{pred}}=\sigma_{x}\sqrt{1\;-\;r_{xx\prime }^{2}}$$

#### 2.5.2. Neurological Disability Worsening

Patients with no relapsing activity and no EDSS worsening over the one-year follow-up were defined as neurologically stable. 

#### 2.5.3. Patients’ Subgroups Based on Presence of Cognitive Decline and Neurological Disease Activity

Based on neuropsychological and neurological evaluation over the one-year follow-up as described above, we divided our sample into four groups: Group 1: Both neurologically and cognitively stable, Group 2: Neurologically stable but cognitively worsening (analogous to the basic concept of ICD), Group 3: Cognitively stable but neurologically worsening, Group 4: Both neurologically and cognitively worsening over the one-year follow-up. 

To verify the relevance of the concept of ICD we additionally applied strict ICD evidence criteria with additional information about MRI activity, and possible comorbidities, such as depression. Therefore, the presence of active T2 lesions on brain MRI was investigated in Group 2. In addition, patients were evaluated as depressed when the BDI total score in any of the timepoints was ≥10 [23]. Depressive symptoms worsening was defined as an individual’s score worsening using Lord and Novick RCI [24,25]. 

### 2.6. Statistical Analysis

All analyses were performed using the jamovi 1.0 (www.jamovi.org, accessed on 7 March 2021) and R statistical software (http://www.R-project.org, accessed on 7 March 2021). In case of missing data, we made the pairwise deletion (available-case analysis) to minimize the loss of cases. We used parametric and non-parametric tests, depending on whether variables were normally distributed. To assess the normality of distributions, we used visual inspection of the histograms, inspection of the Q-Q plots, and the Shapiro–Wilk test. 

We tested between-group differences using the One-Way ANOVA (Fischer’s), Kruskal–Wallis H test, and the Chi-squared test based on the variable and distribution type. We used Levene’s test to assess the presumption of equal variances. Post-hoc pairwise comparisons were done using the Dwass–Steel–Critchlow–Fligner test in case of the Kruskal–Wallis H tests, by Tukey’s HSD in case of Fischer’s One-Way ANOVA. For post-hoc analysis of the Chi-squared test, we compared column proportions by Z-scores/χ^2^-scores based on Adjusted Standardized Residuals with Bonferonni correction adjusted significance levels. Similarly, effect sizes were analyzed by ε^2^, η^2^, or Cramer’s V according to the analysis. 

We assessed predictors of ICD within the neurologically stable group by Binomial Logistic Regression with Group 1 and Group 2 used as dependent variables. We selected the final predictors best describing the ICD occurrence based on previously found between-group differences, controlling for the assumption of no collinearity, and based on the Omnibus Likelihood Ratio test.

The level of tested statistical significance was set to α = 0.05. The Benjamini–Hochberg (BH) procedure with Q = 0.1 was used to control the false discovery rate. Uncorrected *p* values are reported. Associations losing significance after the BH procedure are described as “trends”.

## 3. Results

### 3.1. Description

The median age at baseline was 37.6 years, median disease duration was 8.2 years and 69.6% of patients were females. The median education was 14.0 years. 

### 3.2. Cognitive Worsening over the One-Year Follow-Up

During the one-year follow-up, 727 (66.6%) patients were neurologically stable, and 1021 (93.6%) patients were cognitively stable. When combined, 683 (62.6%) patients remained completely stable with no neurological or cognitive worsening (Group 1). However, 44 neurologically stable patients (Group 2; 4.0% of all patients) deteriorated in the SDMT test and thus experienced ICD (see Figure 2). Similar results were observed in the confirmatory analysis after one year (Appendix A). The cut-off value of SDMT score decrease to evaluate patient’s status as cognitively worsening oscillated around median = −10 points (IQR = −14; −8). 

Out of Group 2 (ICD), 26 patients (59.1% of Group 2 (81.3% if cases with missing MRI data are excluded); 2.4% of all patients) showed concurrent radiological disease activity while cognitively worsening. Over half of the Group 2 patients were depressed (*n* = 24, 54.5%) and most of them had stable or improving depressive course over the follow-up (*n* = 43, 97.7%). Overall, 34.1% (*n* = 15) (46.9% if cases with missing MRI data are excluded) of individuals from Group 2 met the strict ICD criteria (concurrent MRI disease activity and no depressive symptomatology) indicating that 1.4% of our total sample experienced confirmed ICD in SDMT over the one-year follow-up when the strict criteria were applied (Figure 2). 

### 3.3. Differences between Cognitively Stable and Cognitively Worsening Patients with or without Clinical Disease Activity at Baseline and Follow-Up

The characteristics of the four patient groups, defined by their neurological and cognitive status over the one-year follow-up, differed in almost all neurological, neuropsychological and radiological baseline measures. However, the effect sizes of the differences were low, usually below 0.1 (Table 1 and Table 2, Appendix A). 

The cognitively and neurologically worsening patients (Group 4) performed worse in important radiological measures, compared to cognitively stable groups (1 and 3). Group 4 patients had higher baseline T1 and T2 lesion load and lower BPF and CCF scores. Group 4 did not differ in these markers from Group 2 (ICD). Alongside no difference between Group 2 (ICD) and the cognitively stable Groups 1 and 3, it placed the performance of Group 2 (ICD) in between Group 4 and both cognitively stable groups. This pattern changed slightly in the confirmatory analysis where Group 2 (ICD) switched its place with Group 4. Generally, cognitively worsening groups showed a trend to score worse than cognitively stable groups on a majority of important disease severity markers (Table 2 and Appendix A). This is well illustrated also in Figure 3 and Appendix A, where we clearly see the trend of both cognitively worsening groups to score worse than the cognitively stable groups in major neurological and radiological baseline measures.

Completely stable patients from the Group 1 had the smallest proportion of depressive symptoms (*n* = 232; 34.0%; z = 4.57, *p* < 0.001). Group 4 had higher prevalence of cognitive impairment at the baseline (Group 4: z = 3.09, *p* = 0.002), compared to Group 1 and Group 2. Proportions of cognitively impaired patients at baseline in different subgroups were as follows: Group 1 = 29.9%, Group 2 = 27.3%, Group 3 = 39.6%, and Group 4 = 61.5%) (Table 1 and Table 2 and Figure 3).

### 3.4. Predictors of Cognitive Decline

In the binary logistic model, we found only trends for the association between ICD and investigated predictors such as EDSS and BPF at baseline (*p* between 0.034 and 0.069) or BPF and SDMT Z-Score at Month 12 in the confirmatory analysis (*p* between 0.003 and 0.004). The percentages of total variance explained by investigated predictors were relatively low and the results were not confirmed in the confirmatory analysis (Appendix A).

More than one-third of the patients (*n* = 28; 40.0%) who experienced cognitive decline during the one-year follow-up were already cognitively impaired at the baseline. When analyzed only patients with normal cognitive performance at baseline, most of the between-group differences (between Groups 1 to 4) disappeared. We did not find any strong predictor of an upcoming cognitive decline. The between-group differences found in the primary analyses (Table 1 and Table 2) were caused mainly by the persons who were already cognitively impaired (Appendix A).

## 4. Discussion

### 4.1. What Is the Proportion of Cognitively Worsening Patients?

In this large sample study, we showed that annual cognitive screening by processing speed test SDMT can detect cognitive decline in 6.4% of MS patients. Together, 4.0% of patients experienced cognitive decline while being neurologically stable (ICD) and 1.4% of patients experienced cognitive decline while being neurologically stable but having radiological disease activity and no depressive symptomatology (i.e., met the strict ICD criteria). Indeed, only around 0.6% of patients experienced cognitive worsening unexplainable by clinical or radiological disease activity (i.e., radiologically unconfirmed ICD). Given that the annual cognitive screening of all patients is a resource-demanding activity [2,29], these results show that a very basic cognitive screening assessment, such as SDMT, has the potential to additionally detect cognitive symptoms worsening in 6% of our patients. Importantly, in proportion of these patients (0.6%), disease activity as detected by cognitive decline was undetectable by other common clinical or imaging monitoring tools. 

By combining the results of the main and confirmatory analyses, we can conclude that cognitively worsening patients (Groups 2 and 4) showed a trend to perform worse in almost all neurological, radiological, and neuropsychological baseline measures. However, the effect sizes of the mean differences between the groups were low and the differences disappeared in people with normal cognitive performance at the baseline. 

### 4.2. Recommendations for Daily Practice

#### 4.2.1. Evaluation of Meaningful Change

The annual screening opens various reliability issues [9]. The diagnosis of cognitive decline is naturally influenced by the technique used to evaluate the meaningful change in scores. The proportion of the 4.0% of patients with ICD is based on RCI with a conservative 95% CI [24,25]. When compared to the previously suggested clinically meaningful 4-point decline in SDMT [30,31], our approach is probably highly specific, and minimizing Type I Error may underestimate the real proportion of patients with cognitive decline. The RCI provides a conservative estimate on the proportion of people experiencing ICD, accounting for reliability and random error of measurement issues and thus it can be used for reliable clinical evaluation purposes [9,24]. The large proportion of ICD validated by concurrent radiological activity (i.e., 81%) supports this view. Nevertheless, future research should define proportions and characteristics of patients with ICD when defined by the previously suggested clinically meaningful 4-point decline in SDMT and compare them with our results and other RCI methodologies. 

#### 4.2.2. Pros and Cons of the Annual Screening

The annual cognitive screening was previously recommended as the standard of cognitive functions monitoring, but opinions on whether it should be used for decision-making on treatment changes differ [2,8,9]. Our findings suggest that in most cases, the cognitive changes are associated with other clinical or radiological disease-activity markers. Cognitive monitoring can certainly improve our disease activity monitoring and thus decision-making. Even in early/mild/preclinical forms of MS, the cognitive difficulties are relatively frequent [32], represent a risk factor for future disease course [33], and in patients with low cognitive reserve, even small structural damages can result in irreversible cognitive deterioration [7,11,16]. Therefore, it is essential to detect the ongoing disease activity as soon as possible to intervene therapeutically. Patients with cognitive worsening (Groups 2 and 4) had a trend for higher disease burden at baseline than patients with stable cognition over the follow-up (Groups 1 and 3). Also, the cognitive decline did not affect the change of the total EDSS score. It corresponds to a general trend where the cognitive outcomes are clinically underestimated [34]. That can be harmful to the patient if the disease activity continues undetected [7,11,33]. Furthermore, more sophisticated nonconventional MRI techniques may be needed to detect imaging correlates of disease activity associated with cognitive worsening [10,12,13,35], thus cognitive monitoring could hypothetically serve as an easy proxy for disease activity undetectable by conventional MRI.

In a previous study, we suggested integrated MRI measures [15] which could help to select patients who may benefit from the cognitive assessment the most. If we integrate it with our current findings, we assume that although those with the advanced MRI pathology, such as high lesion volume and advanced brain atrophy have the highest probability of cognitive decline [15], in people with less severe MS the cognitive decline might be the only measure suggesting an ongoing disease activity, when only conventional MRI measures are applied.

### 4.3. Limitations

The study includes three consecutive time points, although four time points would be necessary to fully understand the personal longitudinal cognitive profile by allowing to set up practice and confirmation phases. Future analyses evaluating the effectivity of various screening frequencies would be highly beneficial.

Our RCI evaluation is based on a preliminary normative sample from a validation study [19], adequate local norms are yet to be published, however with the conservative CI applied, we do believe the potential bias was minimized. To assess depressive symptoms, the BDI was used [23], the somatic symptoms of depression from this scale overlap with some symptoms of MS and thus it may lead to an increased amount of false-positive cases. Cognitive functions were assessed by the SDMT. While this test evaluates the most frequently impaired cognitive domain in MS, the processing speed, several domains such as episodic memory, higher executive functions, visual-spatial ability, or phonemic fluency were not analyzed [3,19]. While applying strict ICD criteria, we would like to point out that the MRI activity and depressive symptomatology do not necessarily relate to the detected ICD. The assessment of neither of these two symptoms is necessary for the clinical relevance of ICD per se but both measures can give us insight into ICD circumstances. MRI activity illustrates whether there was ongoing concurrent radiological disease activity [1] and depressive symptomatology refers to other possible sources of cognitive changes [36], but the relationship is not necessarily causal and serves only for illustrational purposes. Although we enrolled a large sample, our sample consists of predominantly patients with a short disease duration and low disease burden, thus probably with lower cognitive impairment. Despite our large sample size, the groups representing patients with cognitive decline are inadequately small to present definitive answers. Considering our initial large sample size, more rigorous analyzes would require a multicentric coordinated approach.

To make the recommended annual testing more sensitive while keeping it reliable, there is still enough work to be done. Data on normative longitudinal trajectories are essential [37] and still not available for various tests, not to mention the big potential of more sensitive tests based on the up-to-date theoretical framework [1,32,38], or computer-based ecologically-valid testing based on smart-device data collection [39,40,41].

## 5. Conclusions

In conclusion, during the one-year follow-up, a small proportion of MS patients experienced isolated cognitive decline detectable by rigorous criteria applicable in daily clinical practice. The majority of the patients with isolated cognitive decline showed concurrent MRI activity. Thus, cognitive screening by SDMT test can provide clinicians new information about ongoing MS symptoms they would miss otherwise. Patients with severe MS are more prone to cognitive decline, however, those with healthy cognition and mild MS might benefit from the early detection of cognitive changes the most.

## Figures and Tables

**Figure 1 diagnostics-11-00464-f001:**
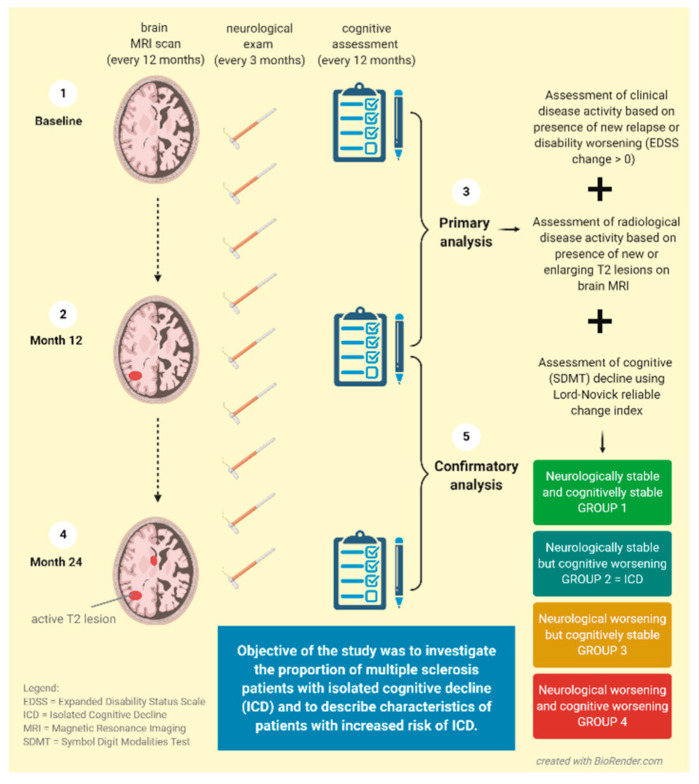
Design of the Study and Analysis.

**Figure 2 diagnostics-11-00464-f002:**
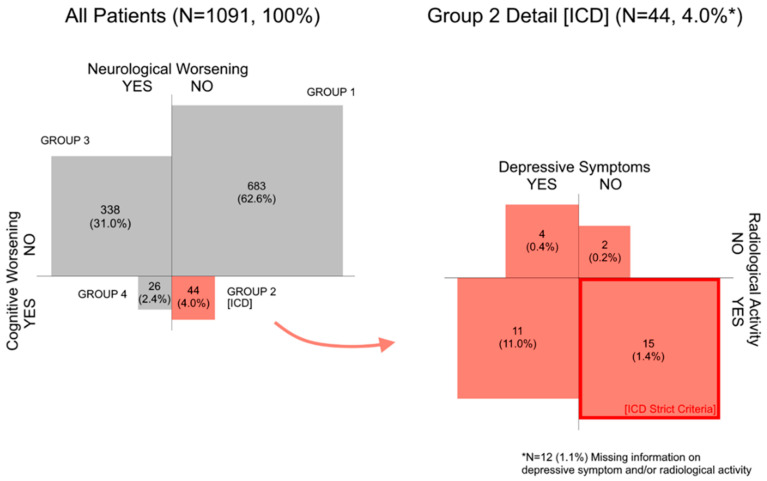
Neuropsychological and Neurological Evaluation between Baseline and Month 12.

**Figure 3 diagnostics-11-00464-f003:**
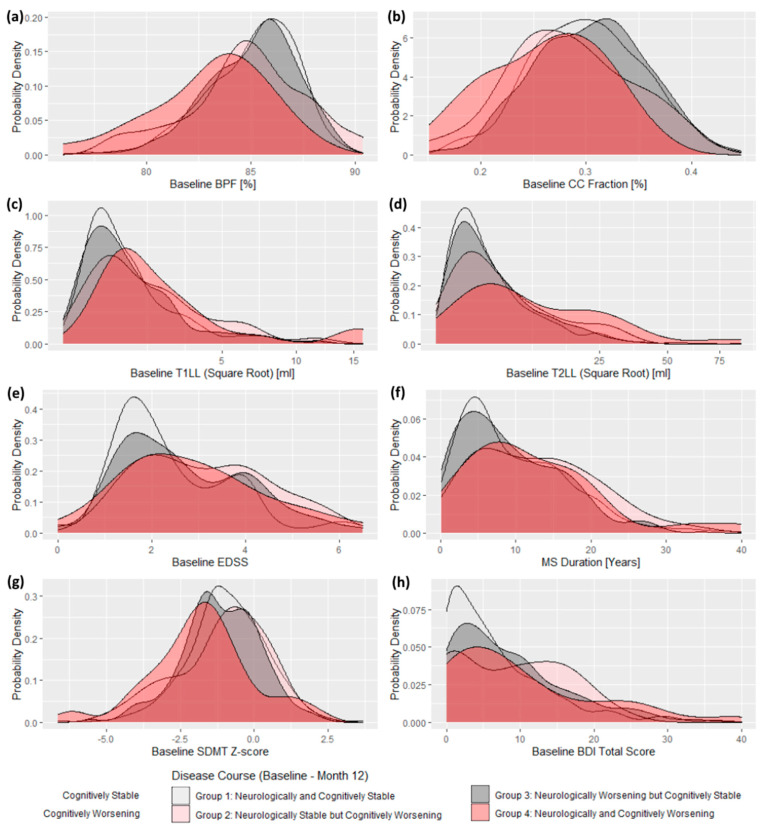
Between-group differences in baseline radiological (**a**–**d**), neurological (**e**,**f**), and neuropsychological (**g**,**h**) scores showed on probability density plots: example of the similar trend of worse baseline disease markers in both cognitively worsening groups (red), in a contrast to both cognitively stable groups (grey). (Legend: BPF = Brain parenchymal fraction, CC = Corpus callosum, T1LL = T1 Lesion Load, T2LL = T2 Lesion Load, EDSS = Expanded Disability Status Scale, SDMT = Symbol Digit Modalities Test, BDI = Beck Depression Inventory.)

**Table 1 diagnostics-11-00464-t001:** Basic sample characteristics at baseline.

Characteristics	*n* Total	All Groups	GROUP 1	GROUP 2	GROUP 3	GROUP 4	Sig.	Effect Size
Demographic								
*n* of patients (%)	1091	1091 (100%)	683 (62.6%)	44 (4.0%)	338 (31.0%)	26 (2.4%)	*n*/A	*n*/A
Females (%)	1091	759 (69.6%)	475 (69.6%)	25 (56.8%)	244 (72.2%)	15 (57.7%)	0.102 ^1^	0.075 ^2^
Age	1091	38.4 ± 9.0	38.6 ± 9.2	40.1 ± 8.4	37.6 ± 8.7	40.3 ± 9.3	0.113 ^3^	0.006 ^4^
Education	1091	14.7 ± 3.0	14.9 ± 3.0	14.1 ± 2.7	14.6 ± 3.0	15.1 ± 3.8	0.148 ^3^	0.005 ^4^
Neurological								
Disease Duration	1090	9.9 ± 7.3	9.9 ± 7.3	12.7 ± 7.9	9.4 ± 6.8	12.4 ± 9.3	**0.028 ^3^**	0.008 ^4^
EDSS†	1091	2.0 (1.5–3.5)	2.0 (1.5–3.5)	3.0 (2.0–4.0)	2.5 (1.5–4.0)	2.75 (2.0–4.0)	**<0.001 ^3^**	0.019 ^4^
DMT (%)	1091						0.866 ^1^	0.034 ^2^
High Efficacy		175 (16.0%)	109 (16.0%)	9 (20.5%)	52 (15.4%)	5 (19.2%)		
Low Efficacy		766 (70.2%)	482 (70.6%)	31 (70.5%)	237 (70.1%)	16 (61.5%)		
Neuropsychological								
Cogn. Impaired ‡ (%)	1091	366 (33.5%)	204 (29.9%)	12 (27.3%)	134 (39.6%)	16 (61.5%)	**<0.001 ^1^**	0.134 ^2^
SDMT Z-Score	1091	−1.0 ± 1.36	−0.92 ± 1.35	−1.07 ± 1.53	−1.13 ± 1.32	−1.77 ± 1.74	**0.004 ^3^**	0.012 ^4^
BDI Total Score	1091	7.6 ± 7.4	6.8 ± 7.0	8.8 ± 7.3	8.7±7.7	10.0 ± 10.2	**<0.001 ^3^**	0.018 ^4^
Radiological								
BPF (%)	969	85.0 ± 2.24	85.1 ± 2.16	84.8 ± 2.84	85.1 ± 2.19	83.1 ± 2.83	**0.005 ^3^**	0.013 ^4^
CCF (%)	968	0.30 ± 0.05	0.30 ± 0.05	0.28 ± 0.06	0.30 ± 0.05	0.26 ± 0.06	**<0.001 ^5^**	0.017 ^6^
T1 Lesion Volume (mL)	944	1.59 ± 2.02	1.50 ± 1.87	2.18 ± 2.44	1.57 ± 1.96	3.01 ± 4.11	**0.007 ^3^**	0.013 ^4^
T2 Lesion Volume (mL)	944	4.93 ± 8.31	4.68 ± 8.35	6.17 ± 8.35	4.63 ± 6.56	13.0 ± 18.0	**0.007 ^3^**	0.013 ^4^

Legend: Group 1 = Neurologically and Cognitively Stable, Group 2 = Neurologically Stable but Cognitively Worsening, Group 3 = Neurologically Worsening but Cognitively Stable, Group 4 = Neurologically and Cognitively Worsening, DMT = Disease modifying treatment, Cogn. Impaired = Cognitively Impaired, BPF = Brain parenchymal fraction, CCF = corpus callosum fraction; DMT = disease modifying treatment; ^1^ χ^2^-test, ^2^ Cramer’s V, ^3^ Kruskal–Wallis H test, ^4^ ε^2^, ^5^ Fischer’s One-Way ANOVA, ^6^ η^2^; Differences losing significance after BH procedure are not highlighted. Unless otherwise indicated, mean ± SD reported. † Reported median and interquartile range. ‡ Cognitive impairment is defined as SDMT below the benchmark of −1.5 SD, using norms adjusted for age and education.

**Table 2 diagnostics-11-00464-t002:** Post-Hoc pairwise comparison of selected subgroups at baseline.

	GROUP 1 vs. GROUP 2		GROUP 1 vs. GROUP 4		GROUP 2 vs. GROUP 4		GROUP 3 vs. GROUP 4	
	Mean Difference/W	Sig.	Mean Difference/W	Sig.	Mean Difference/W	Sig.	Mean Difference/W	Sig.
Disease Duration	3.39	0.078 ^1^	1.98	0.499 ^1^	−0.62	0.972 ^1^	2.35	0.346 ^1^
EDSS	4.91	**0.003** ^1^	2.28	0.371 ^1^	−1.15	0.848 ^1^	0.76	0.950 ^1^
SDMT Z-Score	−0.19	0.999 ^1^	−4.13	**0.018** ^1^	−2.79	0.197 ^1^	−3.17	0.112 ^1^
BDI Total Score	2.29	0.370 ^1^	2.06	0.465 ^1^	0.31	0.996 ^1^	0.19	0.999 ^1^
BPF (%)	−0.94	0.911 ^1^	−5.05	**0.002** ^1^	−3.05	0.136 ^1^	−4.82	**0.004** ^1^
CCF (%)	0.02	0.232 ^2^	0.04	**0.003** ^2^	0.02	0.422 ^2^	0.04	**0.002** ^2^
T1 Lesion Volume (mL)	2.70	0.225 ^1^	4.30	**0.013** ^1^	1.53	0.702 ^1^	3.90	**0.030** ^1^
T2 Lesion Volume (mL)	1.69	0.630 ^1^	4.69	**0.005** ^1^	2.56	0.267 ^1^	4.28	**0.013** ^1^
BPF Change (%)	−3.03	0.141 ^1^	−4.18	**0.017** ^1^	−2.07	0.461 ^1^	−3.35	0.083 ^1^

Legend: Group 1 = Neurologically and Cognitively Stable, Group 2 = Neurologically Stable but Cognitively Worsening, Group 3 = Neurologically Worsening but Cognitively Stable, Group 4 = Neurologically and Cognitively Worsening, BPF = Brain parenchymal fraction, CCF = corpus callosum fraction; ^1^ Dwass–Steel–Critchlow–Fligner, ^2^ Tukey. Differences losing significance after BH procedure are not highlighted.

## Data Availability

The data presented in this study are available on request from the corresponding author. The data are not publicly available due to privacy restrictions.

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
