# Peer review of "Isolated Cognitive Decline in Neurologically Stable Patients with Multiple Sclerosis"

_diagnostics, 2021, doi:10.3390/diagnostics11030464_

Round 1

Reviewer 1 Report

The authors provided clear answers to the reviewers' concerns and the manuscript has been improved.

Minor points: Figure 3 is still difficult to read. The colors representing groups 2 and 4 are very similar and it is difficult to differentiate between them. The meaning of the abbreviations that appear in the x-axis legend should be explained by a footnote or in the legend.

Author Response

The authors provided clear answers to the reviewers' concerns and the manuscript has been improved.

We thank the reviewer for the previous suggestions which helped us to considerably improve our paper.

Minor points: Figure 3 is still difficult to read. The colors representing groups 2 and 4 are very similar and it is difficult to differentiate between them. The meaning of the abbreviations that appear in the x-axis legend should be explained by a footnote or in the legend.

We thank the reviewer for this comment. We have added legend to the Figure 3, explaining all the abreviations used in the x-axis. We have also slightly changed the colouring to improve the readability of the whole figure.

Reviewer 2 Report

The paper is well written and interesting, I have not any other comments for the authors.

Author Response

The paper is well written and interesting, I have not any other comments for the authors.

We thank the reviewer for this positive report.

This manuscript is a resubmission of an earlier submission. The following is a list of the peer review reports and author responses from that submission.

Round 1

Reviewer 1 Report

This manuscript reports and analyzes changes in cognitive function in multiple sclerosis patients studied in a longitudinal cohort study.

The study has several strengths: a large sample size, well characterized patients for neurological disease, and longitudinal follow-up.

Despite these positive aspects, the study has serious methodological flaws that make it unsuitable for publication in the journal.

The main objective of the study is to determine the proportion of patients who develop "isolated cognitive decline". The answer to the main objective  is not given clearly. The ICD is not clearly defined. In addition, it appears that 33% of participants had cognitive impairment at baseline (Table 1).  Many results are based on the number of patients with worsening cognition, but this is not clearly defined. The authors give complex formulas but the abbreviations included in these formulas are not explained and no reader can understand how these formulas apply to the date recorded in the study.

The authors said (L145) that “ICD evidence criteria require …  information about MRI activity” This point is surprising because a rich literature had established the weak correlation between brain imaging and cognition. The authors should provide more convincing evidence explaining how MRI activity can be taken into account to define cognitive decline.

The study design is very confusing. The authors define 4 groups based on one-year changes in neurological and neuropsychological assessments. And their analysis presented in Table 2 showed that there are significant differences between the groups in the neurological and neuropsychological scores used in the study. This is the king of tautological design!

Figure 3 is also confusing because the Y-axis values are not defined.

Reviewer 2 Report

I carefully read the paper ‘Isolated cognitive decline in neurologically stable patients with multiple sclerosis’. In my opinion, it is an interesting paper, well written and I have not main criticisms. I have only a minor observation about figure 3 that is not easy to understand. Hence, I suggest to improve the description of figure 3 in the caption and describe more extensively his meaning in the results and in the discussion.

With my best regards.